# The Unreliability of Explanations in Few-shot Prompting for Textual Reasoning

**Xi Ye**      **Greg Durrett**
Department of Computer Science
The University of Texas at Austin
`{xiye,gdurrett}@cs.utexas.edu`

## Abstract

Does prompting a large language model (LLM) like GPT-3 with explanations improve in-context learning? We study this question on two NLP tasks that involve reasoning over text, namely question answering and natural language inference. We test the performance of four LLMs on three textual reasoning datasets using prompts that include explanations in multiple different styles. For these tasks, we find that including explanations in the prompts for OPT, GPT-3 (davinci), and InstructGPT (text-davinci-001) only yields small to moderate accuracy improvements over standard few-show learning. However, text-davinci-002 is able to benefit more substantially.

We further show that explanations generated by the LLMs may not entail the models' predictions nor be factually grounded in the input, even on simple tasks with extractive explanations. However, these flawed explanations can still be useful as a way to verify LLMs' predictions post-hoc. Through analysis in our three settings, we show that explanations judged by humans to be good—logically consistent with the input and the prediction—more likely cooccur with accurate predictions. Following these observations, we train calibrators using automatically extracted scores that assess the reliability of explanations, allowing us to improve performance post-hoc across all of our datasets.[1]

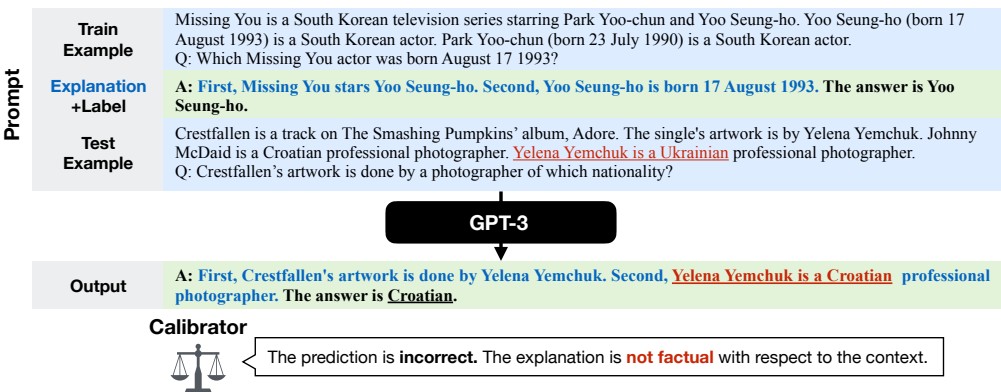

Figure 1: Prompting GPT-3 with explanations. By including explanations in the in-context examples, we can cause GPT-3 to generate an explanation for the test example as well. In this case, the generated explanation is nonfactual, despite the simple reasoning involved here. However, we show this nonfactuality actually provides a signal that can help calibrate the model.

---

[1]Data and code available at `https://github.com/xiye17/TextualExplInContext`

36th Conference on Neural Information Processing Systems (NeurIPS 2022).

# 1    Introduction

Recent scaling of pre-training has empowered large language models (LLMs) to learn NLP tasks from just a few training examples "in context," without updating the model's parameters (Brown et al., 2020). However, this learning process is still poorly understood: models are biased by the order of in-context examples (Zhao et al., 2021) and may not leverage the instructions or even the labels of the examples in the ways one expects (Min et al., 2022; Webson and Pavlick, 2022). Existing tools for interpreting model predictions have high computational cost (Ribeiro et al., 2016) or require access to gradients (Simonyan et al., 2014; Sundararajan et al., 2017), making them unsuitable for investigating in-context learning or explaining the predictions of prompted models.

One appealing way to gain more insight into predictions obtained through in-context learning is to let the language model "explain itself" (Nye et al., 2021; Wei et al., 2022; Chowdhery et al., 2022; Marasović et al., 2022; Lampinen et al., 2022). In addition to input-label training pairs in context, one can prompt the language model with an explanation for each pair and trigger the model to generate an explanation for its prediction (Figure 1). Prompting with explanations introduces much richer information compared to using labels alone, which might guide the inference process and allow the model to learn more information from the examples.

In this work, we investigate the nature of the explanations that LLMs generate and whether they can improve few-shot in-context learning for textual reasoning tasks, specifically QA and NLI. Recent prior work that finds success with this approach largely targets symbolic reasoning tasks with a very different structure, such as math word problem solving (Nye et al., 2021; Wei et al., 2022). We experiment on three different datasets spanning QA and NLI with four LLMs: OPT, GPT-3 (davinci), InstructGPT (text-davinci-001), and text-davinci-002. The results suggest that explanations only substantially improve accuracy for text-davinci-002, but give a smaller improvement or even hurt the performance with the other LLMs.

Surprisingly, we find that the explanations generated by LLMs can be **unreliable**, even for a very simple synthetic dataset. We evaluate the explanations along two axes: *factuality*, whether the explanation is correctly grounded in the input, and *consistency*, whether the explanation entails the final prediction. LLMs tend to generate consistent explanations that account for the predictions, but the explanations may not be factual, as as shown in Figure 1. Furthermore, our analysis suggests an unreliable explanation more likely indicates a wrong prediction compared to a reliable explanation.

Despite LLMs' failures here, we can still benefit from model-generated explanations by using them for calibration. If we are able to automatically assess the reliability of an explanation, we can allow an LLM to return a null answer when its explanation is unreliable, since the prediction in this case is less likely to be correct. Unfortunately, there is no automated way to perfectly assess the reliability, but we can extract features that approximately reflect it. We use these features to calibrate InstructGPT's[2] predictions, and successfully improve the in-context learning performance across all the datasets.

In summary, our main findings are: (1) Simply plugging explanations into the prompt does not always substantially boost the in-context learning performance for textual reasoning. (2) LLMs generate explanations consistent with their predictions, but these explanations might not be factually grounded in the inputs. (3) The factuality of an explanation can serve as an indicator for the correctness of the corresponding prediction. (4) Using features that can approximate the factuality of explanations, we successfully use explanations to improve the in-context learning performance across all tasks.

# 2    Does Prompting with Explanations Improve In-Context Learning?

In this paper, we specifically focus on tasks involving reasoning over natural language. These are tasks where explanations have been traditionally studied (Camburu et al., 2018; Rajani et al., 2019), but which are more complex than tasks like sentiment analysis which are well explained by extractive rationales (Zaidan et al., 2007; DeYoung et al., 2020). We experiment on two tasks,

---

[2]Throughout our paper, we primarily test on InstructGPT for two reasons. First, it was the most capable model available at the time we were conducting the majority of our experiments. Second, it still has significant room to improve on the datasets we explore in this work. This setting is a representative testbed for the situation where an LLM-based system does not yet give satisfactory performance on a target task, causing the system designer to turn to explanations in prompts to improve things.

| | | | |
|---|---|---|---|
| SYNTH | **Context:** | Christopher agrees with Kevin. Tiffany agrees with Matthew. Mary hangs out with Danielle. James hangs out with Thomas. Kevin is a student. Matthew is a plumber. Danielle is a student. Thomas is a plumber. | |
| | **Question:** | Who hangs out with a student? | |
| | **Answer:** | Mary    **Explanation:** Danielle is a student and Mary hangs out with Danielle. | |
| E-SNLI | **Premise:** | A toddler in a green jersey is being followed by a wheelchair bound woman in a red sweater past a wooden bench. | |
| | **Hypothesis:** | A toddler is walking near his wheelchair bound grandmother. | |
| | **Label:** | Neither    **Explanation:** the woman may not be his grandmother. | |

Figure 2: A SYNTH example and an E-SNLI example. See Figure 3 for ADVHOTPOT examples.

reading comprehension question answering (QA) and natural language inference (NLI), on three English-language datasets. For each dataset, we create a test set with 250 examples.

## 2.1 Datasets

**Synthetic Multi-hop QA (SYNTH)**   In order to have a controlled setting where we can easily understand whether explanations are factual and consistent with the answer, we create a synthetic multi-hop QA dataset. Shown in Figure 2, each example in this dataset asks a bridge question (using the terminology of Yang et al. (2018)) over a context consisting of supporting facts paired with controlled distractors. This dataset is carefully designed to avoid spurious correlations, giving us full understanding over the correct reasoning process and the explanation for every example, which naturally consists of the two supporting sentences. See Appendix B for full details of this dataset.[3]

**Adversarial HotpotQA (ADVHOTPOT)**   We also test on the English-language Adversarial HotpotQA dataset (Yang et al., 2018; Jiang and Bansal, 2019). We use the adversarially augmented version since InstructGPT achieves high performance on the distractor setting of the original dataset. We make a challenging set of examples by balancing sets of questions on which InstructGPT makes correct and incorrect predictions. The context of each question includes two ground truth supporting paragraphs and two adversarial paragraphs. Full details of preprocessing the ADVHOTPOT dataset can be found in Appendix C.

For ADVHOTPOT, we manually annotated explanations for the training examples. Figure 1 shows an example of such an explanation, highlighted in orange. We could use the supporting sentences as the explanations, but we found they are usually too verbose and not sufficient, e.g., with anaphors that resolve outside of the supporting sentences. Therefore, we manually annotate a set of explanations which clearly describe the reasoning path for each question.

**E-SNLI**   E-SNLI (Camburu et al., 2018) is an English-language classification dataset commonly used to study explanations, released under the MIT license. Shown in Figure 2, each example consists of a premise and a hypothesis, and the task is to classify the hypothesis as entailed by, contradicted by, or neutral with respect to the premise. As a notable contrast to the other datasets, the explanations here are more *abstract* natural language written by human annotators, as opposed to mostly constructed from extracted snippets of context.

## 2.2 Baselines

We study the effectiveness of plugging in explanations by comparing the in-context learning performance of prompting with or without explanations. Prompting without explanations resembles the standard few-shot in-context learning approach (**Few-Shot**). To incorporate explanations into the prompt, we consider the following two most commonly used paradigms:

**Explain-then-Predict (E-P)** prepends an explanation before the label (Figure 1). The language model is expected to generate an explanation first followed by the prediction. The prompting style of past work involving computational traces can be categorized into this paradigm, including Nye et al. (2021) and Wei et al. (2022). This approach is also called a pipeline model in other literature on training models using explanations (Jacovi and Goldberg, 2021; Wiegreffe et al., 2021).

---

[3]This dataset is inspired by task 15 of the bAbI dataset (Weston et al., 2016). In our preliminary experiments with some of the other bAbI tasks, we found poor performance from InstructGPT similar to our results on SYNTH, both with and without explanations.

Table 1: Results of prompting with explanations on four large language models. Using explanations leads to small to moderate improves performance on OPT, GPT-3, and InstructGPT, and has more prominent effects on text-davinci-002.

| | | SYNTH | ADVHOTPOT | E-SNLI |
|---|---|---|---|---|
| OPT (175B) | FEW-SHOT | **40.5**$_{2.8}$ | 49.7$_{2.6}$ | **44.0**$_{3.8}$ |
| | E-P | 29.6$_{0.5}$ | **52.6**$_{6.5}$ | 39.3$_{7.8}$ |
| | P-E | 40.2$_{2.6}$ | 43.3$_{4.5}$ | 43.4$_{1.6}$ |
| GPT-3 | FEW-SHOT | 49.5$_{0.6}$ | 49.1$_{6.2}$ | 43.3$_{5.7}$ |
| | E-P | 47.1$_{2.8}$ | **54.1**$_{4.1}$ | 40.4$_{4.5}$ |
| | P-E | **51.3**$_{1.8}$ | 48.7$_{4.6}$ | **48.7**$_{2.4}$ |
| InstructGPT | FEW-SHOT | 54.8$_{3.1}$ | 53.2$_{2.3}$ | 56.8$_{2.0}$ |
| | E-P | **58.5**$_{2.1}$ | **58.2**$_{4.1}$ | 41.8$_{2.5}$ |
| | P-E | 53.6$_{1.0}$ | 51.5$_{2.4}$ | **59.4**$_{1.0}$ |
| text-davinci-002 | FEW-SHOT | 72.0$_{1.4}$ | 77.7$_{3.2}$ | 69.1$_{2.0}$ |
| | E-P | **86.9**$_{3.8}$ | **82.4**$_{5.1}$ | **75.6**$_{7.6}$ |
| | P-E | 81.1$_{2.8}$ | 77.2$_{4.8}$ | 69.4$_{5.0}$ |

**Predict-then-Explain (P-E)** generates the explanation after the prediction. Unlike E-P, the predicted explanation does not influence the predicted label, since we use greedy inference and the explanation comes afterwards. However, the explanations in the prompt still impact the predictions.

## 2.3   Setup

For few-shot learning, we use roughly the maximum allowed shots in the prompt that can fit the length limit of OPT (Zhang et al., 2022) and GPT-3 (Brown et al., 2020), which is 16 for SYNTH, 6 for ADVHOTPOT, and 32 for E-SNLI, respectively.[4] We experiment with four LLMs, including OPT (175B), GPT-3 (davinci), InstructGPT (text-davinci-001), and text-davinci-002. OPT and GPT-3 are trained using the standard causal language modeling objective, whereas InstructGPT and text-davinci-002 are trained with special instruction data and human annotations. We generate outputs with greedy decoding (temperature set to be 0). Our prompt formats follow those in Brown et al. (2020). The explanations are inserted before/after the prediction with conjunction words like *because*. Please refer to Appendix A for full prompts. Because the results of in-context learning vary with the examples presented in the input prompt, for each dataset, we randomly sample multiple groups of training shots, and report the mean and standard deviation of the results (subscript). We use 5 groups for InstructGPT, the primary LM we are using throughout our paper, and 3 groups for the rest.

## 2.4   Results

As shown in Table 1, OPT, GPT-3, and InstructGPT show small to moderate improvements from using explanations for textual reasoning tasks. On the two QA tasks, SYNTH and ADVHOTPOT, E-P improves the performance of InstructGPT, the best among these three LMs, from 54.8 to 58.5 and 56.8 to 59.4, respectively.[5] On E-SNLI, P-E outperforms FEW-SHOT by 2.6, whereas E-P substantially lags FEW-SHOT. Comparing E-P against P-E on SYNTH and E-SNLI, E-P typically degrades performance (except on SYNTH for InstructGPT) and P-E is inconsistent across the different models, whereas E-P consistently leads to performance improvements on ADVHOTPOT. There is no single winner between the two paradigms of using explanations; choosing the most effective way is task-specific. Overall, vanilla LLMs (OPT and GPT-3) see limited benefit from producing explanations, and even the Instruct-series InstructGPT does not see substantial improvements.

The only exception is text-davinci-002. text-davinci-002 greatly benefits from explanations in the prompt across all three tasks, and E-P is consistently more effective than P-E. However, it is unclear what contributes to this difference. As far as we are aware, the differences between text-davinci-002

---

[4]This contrasts with recent work like Zhao et al. (2021) that focuses on improving performance in the 1-4-shot setting; by using more data we achieve much stronger results on our tasks.

[5]For SYNTH, we also tried using an alternative style of explanations (reversing the order of the two sentences in the explanations), which leads to mild performance degradation.

| | |
|---|---|
| **Nonfactual** | Pedro Rubens! The individual chapters were published into 64 "tankōbon" by Kodansha.
Yōko Shōji (born 4 June 1950, in Mobara, Chiba) is a Japanese manga artist. She is best known for writing "Seito Shokun!
Mulder Scully! The individual chapters were published into 14 "tankōbon" by Kodansha.
Seito Shokun! The individual chapters were published into 24 "tankōbon" by Kodansha between.
Q: How many chapters does Yōko Shōji's most famous manga have?
A: First, Yōko Shōji's most famous manga is "Seito Shokun!". Second, "Seito Shokun!" has 64 chapters. The answer is 64. |
| **Inconsistent** | Tim Minchin (December 29, 1808 July 31, 1875) was the President of the United States.
Andrew Johnson (December 29, 1808 July 31, 1875) was the President of the United States.
George Andrew Atzerodt (June 12, 1835 – July 7, 1865) was a conspirator, with John Wilkes Booth.
Jesse Andrew Williams (June 12, 1835 – July 7, 1865) was a conspirator, with John Wilkes Booth.
Q: Who was older, George Atzerodt or Andrew Johnson?
A: First, George Atzerodt was born on June 12, 1835. Second, Andrew Johnson was born on December 29, 1808. The answer is George Atzerodt. |

Figure 3: Explanations generated for ADVHOTPOT. InstructGPT may generate nonfactual explanations containing hallucination (red) or inconsistent explanations contradicting the answer (red).

Table 2: Left: factuality (Fac) and consistency (Con) of the generated explanations. Right: the % of the examples whose explanation factuality/consistency is congruent with the prediction accuracy. In general, LLMs tend to generate consistent but less likely factual explanations.

| | | Acc | Fac | Con | Acc=Fac | Acc=Con |
|---|---|---|---|---|---|---|
| | *reliability of explanations generated by InstructGPT* | | | | | |
| InstructGPT | SYNTH (E-P) | 58.4 | 72.8 | 64.8 | 66.5 | 68.8 |
| | SYNTH (P-E) | 54.8 | 51.6 | 95.2 | **89.6** | 57.2 |
| | ADVHP (E-P) | 62.0 | 79.6 | 91.2 | **80.0** | 68.4 |
| | ADVHP (P-E) | 54.0 | 69.2 | 82.0 | **77.6** | 67.2 |
| | E-SNLI (P-E) | 62.0 | — | 98.8 | — | 62.0 |
| | *reliability of explanations generated by other LLMs on* SYNTH | | | | | |
| OPT (175B) | SYNTH (E-P) | 30.0 | 77.2 | 47.2 | 45.6 | 58.8 |
| | SYNTH (P-E) | 39.6 | 64.0 | 81.2 | **69.2** | 49.6 |
| GPT-3 | SYNTH (E-P) | 46.8 | 59.2 | 64.8 | **66.8** | 61.2 |
| | SYNTH (P-E) | 52.4 | 52.4 | 83.2 | **78.4** | 58.0 |
| text-davinci-002 | SYNTH (E-P) | 86.0 | 91.6 | 85.2 | **91.2** | 84.8 |
| | SYNTH (P-E) | 81.6 | 83.2 | 96.4 | **95.8** | 82.8 |

and InstructGPT are not described in any publication or blog post.[6] Comparing GPT-3 and Instruct-GPT, we see the move to Instruct series models is *not* sufficient to explain the difference. Given the lack of transparency with this model, we hesitate to make scientific claims about the results it yields.

Our results do not suggest immediate strong improvements from incorporating explanations across all LLMs, even for our synthetic dataset, contradicting recent prior work. This can be attributed to the difference between the tasks we study. The tasks that receive significant benefits from using explanations in Nye et al. (2021) and Wei et al. (2022) are all program-like (e.g., integer addition and program execution), whereas the tasks in this work emphasize textual reasoning grounded in provided inputs. In fact, in Wei et al. (2022) and Chowdhery et al. (2022), explanations only show mild benefit on open-domain QA tasks like StrategyQA (Geva et al., 2021) that are closer to our setting.

## 3  Can LLMs Generate Factual and Consistent Explanations?

Prompting LLMs with explanations and having models generate them may not guarantee higher performance on our tasks. But what about the quality of the model-generated explanations themselves? We assess the reliability of the explanations for the three datasets, measured in terms of two aspects.

**Factuality** refers to whether a generated explanation is faithfully grounded in the corresponding input context (context for QA and premise/hypothesis pair for NLI). A factual explanation should not contain hallucinations that contradict the context. See Figure 3 for a nonfactual explanation.

**Consistency** measures if the explanation entails the prediction. Our concept of consistency resembles plausibility as described in Jacovi and Goldberg (2021), in that we assess whether the prediction follows from the explanation **as perceived by a human**. See Figure 3 for an inconsistent explanation.

---

[6]One publicly-described difference is the addition of editing and insertion, discussed at `https://openai.com/blog/gpt-3-edit-insert/`, but this does not explain the performance differences we observe.

For SYNTH, we use rules to automatically judge whether an explanation is factual and consistent on all four LLMs. For ADVHOTPOT and E-SNLI, the authors manually inspected the explanations generated by InstructGPT and annotated them for these two characteristics (more details in Appendix D). Note for each setting, the results are based on the explanations and predictions obtained with a single set of training shots. We only show the results of P-E on E-SNLI, as E-P is substantially worse here.

**Results** We summarize the results in Table 2. We only report consistency on E-SNLI, as the explanations for E-SNLI often require some external commonsense knowledge which cannot be easily grounded in the inputs or judged as true or false (examples in Appendix F). The results suggest a disconnect between the model predictions and the "reasoning" in explanations. On InstructGPT, though using explanations improves its performance across three tasks, the generated explanations are *unreliable* (upper section), even for the straightforward synthetic setting. Comparing the factuality of explanations for SYNTH generated by GPT-3, InstructGPT, and text-davinci-002, we see that instruction tuning improves the factuality, but even the most powerful text-davinci-002 still fails to generate explanations that are perfectly grounded in the input context. Overall, LLMs tend to generate consistent explanations (>80% for all three datasets with the right prompt structure), but the explanations are less likely to be factual, which is concerning as they can deceive a user of the system into believing the model's answer.

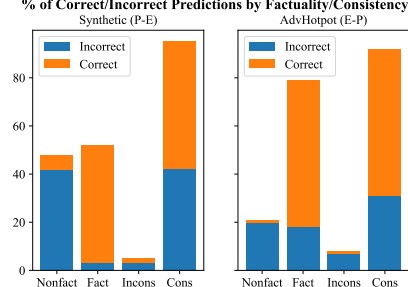

Figure 4: Explanations are more likely to be nonfactual than to be inconsistent, and a nonfactual explanation usually indicates an incorrect prediction.

### 3.1 Reliability of Explanations and Prediction Accuracy

LLMs may hallucinate problematic explanations, but this could actually be advantageous if it gives us a way of spotting when the model's "reasoning" has failed. We investigate the connection between the reliability of an explanation and the accuracy of a prediction and ask whether a reliable explanation indicates an accurate prediction. (This resembles the linguistic calibration of Mielke et al. (2022), but using a different signal for calibration.)

As shown in Table 2 (right), accuracy and factuality/consistency are typically correlated, especially factuality. By knowing whether an explanation is factual, we can guess the model's accuracy a high fraction of the time (Accuracy = Factuality). A nonfactual explanation very likely means an incorrect prediction on the SYNTH dataset across all four LLMs. On ADVHOTPOT, factuality and InstructGPT's prediction correspond 80.0% of the time, substantially surpassing the prediction accuracy itself. We show fractions of correct and incorrect predictions when the explanations are factual/nonfactual and consistent/inconsistent in Figure 4 for two of our settings. Factual explanations are much more likely paired with correct predictions compared to nonfactual explanations. Consistency is also connected to accuracy but is an inferior indicator compared to factuality in general (Table 2).

## 4 Calibrating In-Context Learning using Explanations

From Section 3.1, we see that a human oracle assessment of the factuality of an explanation could be of substantial use for calibrating the corresponding prediction. Can we automate this process?

We first show how to achieve this goal on the perfectly controlled SYNTH dataset (Section 4.1). On our other two datasets, we use surface lexical matching to approximate semantic matching and give real-valued scores approximately reflecting factuality. Following past work on supervised calibration (Kamath et al., 2020; Chen et al., 2021; Ye and Durrett, 2022), we can learn a calibrator that tunes the probabilities of a prediction based on the score of its explanation (Section 4.2). We show such a calibrator can be trained with a handful of examples beyond those used for in-context learning and successfully improve the in-context learning performance on realistic datasets.[7] We note that, as mentioned before, the experiments in this section are conducted on InstructGPT.

---

[7]This procedure does require extra data. However, it provides a natural avenue for using a small number of additional examples that otherwise would be *impossible* to incorporate into this procedure, when the size of the context actually limits the amount of data for in-context learning.

### 4.1 Motivating Example: Improving SYNTH Dataset

We first show how post-hoc calibration functions in the controlled SYNTH setting, where we can simply check the factuality of an explanation. Since the generated explanation always follows the format "`B is [profession] and A [verb] B.`" (example in Figure 2), we can split the explanation into two sentences. The explanation is factual if and only if each of the two sentences exactly matches one of the sentences in the context.

We use the assessment to improve the performance of P-E for SYNTH, where a nonfactual explanation typically indicates an incorrect prediction. This gives us a way to reject presumably incorrect answers. Specifically, we iterate through the top 5 candidate answers (restricted by the API) given by InstructGPT and reject any answer-explanation pair if the explanation is nonfactual until we find a factual one. This procedure dramatically improves the accuracy from 52.4% to 74.8%. Note that this SYNTH dataset is a challenging task given its lack of reasoning shortcuts: for reference, neither ROBERTA (Liu et al., 2019) nor DEBERTA (He et al., 2021) finetuned with 16 examples can achieve an accuracy surpassing 50%. With the help of the explanations and the checking procedure, we can use InstructGPT to achieve strong results using few-shot learning.

### 4.2 Learning-based Calibration Framework

**Framework**   We now introduce the framework that can leverage the factuality assessment of an explanation to calibrate a prediction. Let $p$ be the vector of predicted probabilities associated with each class label in NLI (or the probability score of predicted answer in QA). Let $v$ be a scalar value extracted from the explanation to describe the factuality. Then, we can adjust the probabilities accordingly using a linear model: $\hat{p} = \text{softmax}(W[p; v] + b)$, where $\hat{p}$ is the tuned probabilities.

Our calibration framework is extended from classical calibration methods (Platt, 1999; Guo et al., 2017; Zhao et al., 2021), which apply an affine transformation on the probabilities alone: $\hat{p} = \text{softmax}(Wp + b)$. In contrast, we use an additional factor $v$ in calibration to incorporate the factuality assessment of the explanation.

There are a small number of parameters ($W$ and $b$) that need to be trained in such a calibration framework. We will rely on a few more examples in addition to the shots we use in the prompt to train the calibrator. Specifically, we use the prompt examples to generate the predictions and explanations for these extra examples, and extract predicted probabilities, factors, and target probabilities triples to construct training data points used to train the calibrator. Note this procedure requires **no** explanation annotations for the extra examples.

**Approximating Factuality**   We approximate the factuality using lexical overlap between the explanations and the inputs, which we found to work fairly well for our tasks.

**ADVHOTPOT:** We use an explanation consisting of two sentences (examples in Figure 3) as an illustration. Let $\mathcal{E} = (E^{(1)}, E^{(2)})$ be the generated explanation, where $E^{(1)}$ and $E^{(2)}$ are the two sentences, and the $E^{(i)} = (e_1, e_2, \cdots)$ contain tokens $e_1, e_2, \cdots$. Similarly, let $\mathcal{P} = (P^{(1)}, P^{(2)}, P^{(3)}, P^{(4)})$ be the context paragraphs, and $P^{(i)} = (p_1, p_2, \cdots)$ be the tokens. The factuality estimation of one explanation sentence $E^{(i)}$ is defined as: $\mathcal{V}(E^{(i)}) = \max_{P \in \mathcal{P}} \frac{|E^{(i)} \cap P|}{|E^{(i)}|}$.

Intuitively, the factuality score for a sentence $E$ is defined as the maximum number of overlapping tokens over all paragraphs $P$, normalized by the number of tokens in $E$. We then define the factuality score for the whole explanation as $\mathcal{V}(\mathcal{E}) = \min_{E \in \mathcal{E}} \mathcal{V}(E)$, as it requires all sentences to be factual in order to make the entire explanation factual.[8]

**E-SNLI:** The explanations of E-SNLI do not really involve a concept of factuality. Nevertheless, we use an analogous score following the same principle by viewing the premise as the context. Let $E = (e_1, e_2, \cdots)$ be the explanation and $P = (p_1, p_2, \cdots)$ be the premise. We simply score the explanation by $\mathcal{V}(E) = \frac{|E \cap P|}{|E|}$. The more an explanation overlaps with the premise, the more factual we judge it to be.

---

[8]Alternatively, one might use a fine-tuned NLI model as a proxy (Chen et al., 2021). However, our focus is on the pure black-box setting, and we avoid models that require substantial amounts of data to make work.

## 4.3 Calibrating E-SNLI

**Setup** For E-SNLI, we use calibration methods to postprocess the final probabilities. Unlike classical temperature scaling (Platt, 1999), note that the methods we use here can actually change the prediction; we will therefore evaluate on *accuracy* of the calibrated model.

We study the effectiveness of our explanation-based calibrator under different training data sizes varying from 32 to 128. Recall that we only require explanation annotations for 32 data points, and only need the labels for the rest to train the calibrator. For E-SNLI, we calibrate P-E, which is shown to be more effective than E-P in this setting (Section 2.4).

**Baselines** We provide the performance of fine-tuned RoBERTA (Liu et al., 2019) model as a reference, finding this to work better than De-BERTa (He et al., 2021). To isolate the effec-

Table 3: Accuracy ($\text{mean}_{\text{std dev}}$) of various methods on E-SNLI under different data conditions. **L** denotes number of labels (as well as the total number of examples); **E** denotes the number of explanations. Calibrating using explanations successfully improves the performance of in-context learning.

| w/o Explanation | 32L | 64L | 96L | 128L |
|---|---|---|---|---|
| RoBERTa | $40.1_{4.7}$ | $43.0_{5.1}$ | $49.0_{5.2}$ | $54.9_{4.8}$ |
| FEW-SHOT | $56.8_{2.0}$ | — | — | — |
| FEW-SHOT(NN) | — | — | — | $58.9_{1.0}$ |
| FEW-SHOT+PROBCAL | $61.9_{3.8}$ | $62.4_{2.6}$ | $63.2_{2.9}$ | $63.9_{1.2}$ |
| **w/ Explanation** | **32L+32E** | **64L+32E** | **96L+32E** | **128L+32E** |
| P-E | $59.4_{2.0}$ | — | — | — |
| P-E+PROBCAL | $\mathbf{64.4_{1.8}}$ | $65.4_{1.2}$ | $65.4_{1.6}$ | $65.4_{1.9}$ |
| P-E+EXPLCAL | $64.2_{2.6}$ | $\mathbf{65.8_{1.3}}$ | $\mathbf{67.6_{1.6}}$ | $\mathbf{68.5_{1.2}}$ |
| P-E+ZHANG | $63.0_{3.2}$ | $65.2_{2.2}$ | $65.4_{1.5}$ | $65.9_{2.5}$ |

tiveness of using explanations for calibration, we introduce three additional baselines using non-explanation-based calibrators. We apply the probability-based calibrator as described in Section 4.2 on the results obtained on few-shot learning (FEW-SHOT+PROBCAL) and predict-then-explain pipeline (P-E+PROBCAL). We note that the parameters of these calibrators are trained using the additional data points, as opposed to being heuristically determined as in Zhao et al. (2021). Furthermore, we experiment with a recently proposed supervised calibrator from Zhang et al. (2021), which uses the CLS representations from an additional language model as features in the calibrator. The probabilities are tuned using $\hat{\boldsymbol{p}} = \text{softmax}(W[\boldsymbol{p}; \boldsymbol{h}] + b)$, where $\boldsymbol{h}$ is the CLS representation. Since we do not have access to the embeddings obtained by GPT-3, we use RoBERTA to extract the vectors instead. We use such a calibrator on top of our best-performing base model, P-E, resulting P-E+ ZHANG ET AL. (2021).

Limited by the maximum prompt length, in-context learning is not able to take as input the additional data used for training the calibrator. For a fair comparison, we can allow the in-context model to use this data by varying the prompts across test examples, dynamically choosing the prompt examples to maximize performance. Choosing closer data points for prompting is a common and effective way of scaling up the training data size for in-context learning (Shin et al., 2021; Liu et al., 2021). Following Liu et al. (2021), we test the performance of choosing nearest neighbors for the prompt based on CLS embedding produced by a RoBERTA model (Liu et al., 2019), referred as FEW-SHOT(NN). It is worth clarifying that the FEW-SHOT and FEW-SHOT+PROBCAL approaches use the same set of 32 training shots in the prompt for every test example, whereas the shot sets vary from example to example in FEW-SHOT(NN).

**Results** We show the results in Table 3. We use 5 different groups of training examples and report the mean and standard deviation across the groups. For FEW-SHOT(NN), we only report the results obtained using 128 examples; results using a smaller number of examples will be worse than this.

Under 128 training examples, applying a trained calibrator on top of prompting with explanation (i.e., P-E+EXPLCAL) achieves the best accuracy of 68.5%, which is 12% higher than the performance of the vanilla uncalibrated few-shot in-context learning (FEW-SHOT). P-E+EXPLCAL also outperforms FEW-SHOT+PROBCAL and P-E+PROBCAL by 5% and 3%, respectively. Using explanations is more effective than using probabilities alone. In addition, P-E+EXPLCAL also outperforms P-E+ZHANG ET AL. (2021), whose performance is on par with P-E+PROBCAL. This suggests the additional CLS information is not very helpful in this setting.

As the data size increases from 32 to 128, the performance of the explanation-based calibrator keeps improving notably, whereas the performance of probability-based calibrators nearly saturates at a data size of 96. The performance of FEW-SHOT(NN) with 128 training instances only improves the performance by 3.3%, compared to FEW-SHOT with 32 training instances. Choosing nearest

Table 4: AUC scores (mean$_{std\ dev}$) on ADVHOT-POT under different data conditions. **L** and **E** denotes the number of label annotations and explanation annotations, respectively. Explanation-based calibration successfully improves the performance on top of prompting with explanations.

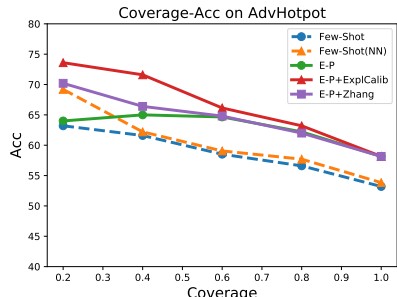

Figure 5: Coverage-Acc curves of various methods on ADVHOTPOT. E-P+EXPLCAL is better calibrated compared to uncalibrated E-P as well as the other approaches.

| w/o Explanation | 6L | 32L | 64L |
|---|---|---|---|
| FEW-SHOT | $59.6_{2.4}$ | — | — |
| FEW-SHOT(NN) | — | — | $61.3_{0.9}$ |
| **w/ Explanation** | **6L+6E** | **32L+6E** | **64L+6E** |
| E-P | $64.4_{2.9}$ | — | — |
| E-P+EXPLCAL | — | $66.0_{3.9}$ | $68.8_{3.0}$ |
| E-P+ZHANG | — | $65.6_{3.9}$ | $66.1_{3.2}$ |

neighbors as the shots, while being effective when having access to a large amount of data, is not helpful in the extreme data-scarce regime. Calibrating using explanations is an effective way of using a few extra data points that cannot fit in the prompt, which is a pitfall of standard in-context learning.

Finally, ROBERTA finetuned using 128 shots only achieves an accuracy of 54.9%, lagging the performance of GPT-3 based models. The limited training data size is insufficient for finetuning smaller language models like ROBERTA, but is sufficient for P-E+EXPLCAL to be effective.

## 4.4 Calibrating ADVHOTPOT

**Setup** For the ADVHOTPOT dataset, our calibration takes the form of tuning the confidence scores of the predicted answers to better align them with the correctness of predictions. These confidence scores can be used in a "selective QA" setting (Kamath et al., 2020), where the model can abstain on a certain fraction of questions where it assigns low confidence to its answers. We use the *area under coverage-accuracy curve* (AUC) to evaluate how well a model is calibrated as in past literature (Kamath et al., 2020; Chen et al., 2021; Zhang et al., 2021; Garg and Moschitti, 2021; Ye and Durrett, 2022). The curve plots the average accuracy with varying fractions (coverage) of questions being answered (examples in Figure 5). For any given coverage, a better calibrated model should be able to identify questions that it performs best on, hence resulting a higher AUC.

We experiment with training data set sizes of 6, 32, and 64. We report the results averaged from 5 trials using different training sets. For ADVHOTPOT, we calibrate E-P, which is shown to be more effective than P-E in this setting (Section 2.4). Our approach is also effective for calibrating P-E; please refer to Appendix E for details.

**Results** We show the AUC scores in Table 4. By leveraging explanations, E-P+EXPLCAL successfully achieves an AUC of 68.8, surpassing both FEW-SHOT by 7 points and E-P by 4 points. We note this is a substantial improvement, given that the upperbound of AUC is constrained by the accuracy of the answers and cannot reach 100. Figure 5 shows the coverage-accuracy curves of various methods averaged across the 5 training runs. E-P+EXPLCAL always achieves a higher accuracy than its uncalibrated counterpart, E-P, under a certain coverage, and the gap is especially large in the most confident intervals (coverage < 50%). E-P+ZHANG ET AL. (2021) is able to calibrate the predictions on this dataset, but still lags our explanation-based calibrator, E-P+EXPLCAL.

In addition, the explanation-based calibrator can be effective with as few as 32 examples. This is because there are only two parameters (the probability of predicted answer and the explanation-based factor) in the calibrator, which can be easily learned in this few-shot setting. Comparing E-P+EXPLCAL against FEW-SHOT(NN), using nearest neighbors in the prompt is also able to improve the performance compared to using a fixed set of shots (FEW-SHOT), yet our lightweight calibrator can better utilize such a small amount of data, and learn to distinguish more accurate predictions based on the explanations.

# 5 Related Work

Our investigation is centered around in-context learning (Brown et al., 2020), which has garnered increasing interest since the breakthrough of various large pretrained language models. Recent work has been devoted to studying different aspects of in-context learning, including its wayward behaviors (Min et al., 2022; Webson and Pavlick, 2022) and approaches to overcome them (Zhao et al., 2021), whereas our exploration focuses on using explanations.

The utility of explanations for few-shot in-context learning has also been discussed concurrently (Nye et al., 2021; Wei et al., 2022; Marasović et al., 2022; Chowdhery et al., 2022; Lampinen et al., 2022; Wiegreffe et al., 2022), especially in symbolic reasoning tasks. We differ in that we study more free-form explanations in tasks (QA and NLI, specifically) focusing on textual reasoning over provided contexts. Furthermore, our work focuses on the nature of the explanations generated by LLMs, which are found to be unreliable. Regarding our use of calibration, similar ideas of explanation-based performance estimation have been applied to other tasks (Rajani and Mooney, 2018; Ye et al., 2021; Ye and Durrett, 2022), but we rely on the free-text explanations generated by the model instead of interpretations obtained through post-hoc interpretation techniques.

More broadly, how to use explanations in various forms (textual explanation, highlights, etc.) to train better models is a longstanding problem (Zaidan et al., 2007). Past work has built a series of pipeline models that first generate the explanations and then make predictions purely based on the generated explanations (Wiegreffe et al., 2021; Zhou and Tan, 2021; Chen et al., 2022). Prior research has also explored using explanations as additional supervision to train joint models (Hancock et al., 2018; Dua et al., 2020; Lamm et al., 2021; Stacey et al., 2022). Another line of work seeks to align the reasoning process of a trained model with the explanations, which is typically done by interpreting a prediction post-hoc through explanation techniques and optimizing the distance between the obtained explanation and ground truth explanation (Liu and Avci, 2019; Rieger et al., 2020; Plumb et al., 2020; Erion et al., 2021; Yao et al., 2021). These aforementioned methods all update the model parameters and typically require a considerable amount of explanation annotations to be effective. By contrast, our setting treats language models as pure black boxes and only requires few-shot explanations.

# 6 Discussion & Conclusion

**Caveats and Risks of Explanations from Large Language Models**   Our analysis suggests that LLMs' internal "reasoning" does not always align with explanations that it generates, as shown by our consistency results. More concerning, the explanations might not be factually grounded in the provided prompt. This shortcoming should caution against any deployment of this technology in practice: because the explanations are grammatical English and look very convincing, they may deceive users into believing the system's responses even when those responses are incorrect. Section 6 of Bender et al. (2021) discusses these risks in additional detail. The fact that language models can hallucinate explanations is also found in other work (Zhou and Tan, 2021). This result is unsurprising in some sense: without sufficient supervision or grounding, language models do not learn meaning as distinct from form (Bender and Koller, 2020), so we should not expect their explanations to be strongly grounded.

We have shown that even explanations which don't lead to accuracy gains can still be useful for calibration. However, the lexical overlap feature we use here is a weak signal of explanation correctness (see the example in Figure 1). Strong enough entailment models should theoretically be able to perform this task and work across a range of tasks without fine-tuning. This explanation assessment model can even be a language model itself trained for this particular propose to approach the verification tasks for a given domain by in-context learning.

**Conclusion**   We have explored the capabilities of LLMs in using explanations in in-context learning for textual reasoning. Through our experiments with four LLMs and on two QA datasets and an NLI dataset, we find that simply including explanations in the prompt does not always improve the performance of in-context learning. Our manual analysis demonstrates that LLMs tend to generate nonfactual explanations when making wrong predictions, which can be a useful leverage to assess the correctness of the predictions. Lastly, we showcase how to use explanations to build lightweight calibrators, which successfully improve InstructGPT's in-context learning performance across all three datasets.

## Acknowledgments

We would like to thank Eunsol Choi, Ruiqi Zhong, Jocelyn Chen, Zayne Sprague, and Jiacheng Xu for their helpful feedback on drafts of this work, as well as the anonymous reviewers for their thoughtful reviews. This work was partially supported by NSF Grant IIS-1814522, NSF CAREER Award IIS-2145280, a grant from Open Philanthropy, a gift from Salesforce Inc., and a gift from Adobe.

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
