# OpenReview forum: "The Unreliability of Explanations in Few-shot Prompting for Textual Reasoning"
_NeurIPS.cc/2022/Conference — NeurIPS 2022 Accept_

### Official Review · Reviewer_5YVG · 2022-07-04

**Rating:** 6
**Confidence:** 3
**Soundness:** 3 good
**Presentation:** 3 good
**Contribution:** 3 good

**Summary:**

This paper explored the capabilities of GPT-3 in using explanations in in-context learning. Although using in-context explanation has been demonstrated quite helpful for symbolic reasoning tasks, this paper finds that it's not the case for textual reasoning tasks such as NLI. Upon investigation, they find that simply including an explanation as in-context example does not always yield big improvements. Their analysis shows that this is because the explanations generated by GPT3 could be nonfactual. Given those observations, they showcase how to use in-context explanation to calibrate GPT3's prediction. The proposed method achieves significant improvements over three datasets.

**Questions:**

Why there are so many missing cells in Table3&4? Will those results affect the conclusion?

**Ethics Review Area:**

["I don’t know"]

**Limitations:**

The authors did not address the limitations and potential negative societal impact of their work.

**Strengths And Weaknesses:**

The observation in this paper is interesting and could be a good complement to current in-context explanation research.
The proposed explanation-based calibration method is simple yet effective.
The paper is clear and easy to follow.

However, I also have the following concerns. The claim that incorporating explanation as in-context examples only marginally improves performance is a strong claim that contradicts prior works. To back up the claim, I suggest the author include a few more datasets in table 1. Besides, for experiments in S2, only 250 testing examples are considered. Can authors justify their choice, as compared to using the official test set? Since on such a small test set, the conclusion is not that convincing.
Besides, for results in Table 1, I do not agree that such improvements (10% relative improvements) are "mild". They are just not that surprising compared to some other tasks.

---

> ### Author Response · Authors · 2022-08-02
> **Thanks for the questions; please find answers below**
>
> **Q:** The claim that incorporating explanation as in-context examples only marginally improves performance is a strong claim that contradicts prior work.
>
> **A:** Please see the general response about framing.
>
> ---
>
> **Q:** Why are there so many missing cells in Table3&4? Will those results affect the conclusion?
>
> **A:** Please see the general response.
>
> ---
>
> **Q:** Only 250 testing examples are considered. Can authors justify their choice, as compared to using the official test set?
>
> **A:** In our preliminary experiments we found that a test set of 250 examples is enough to yield statistically significant differences between the models we examine. Using larger datasets would significantly increase the cost of this work. We instead focused our experimentation on investigating the effects of choosing different “shots”, running multiple trials for each experiment instead of running on more instances per trial. We also include information about cost in Appendix L.
>
> ---
>
> **Q:** For results in Table 1, I do not agree that such improvements (10% relative improvements) are "mild". They are just not that surprising compared to some other tasks.
>
> **A:** Thanks for pointing this out. When assessing the scale of the improvements and choosing to describe them as “mild” or “not substantial”, we are using as calibration the facts that (a) Synth is a synthetic dataset, easily solved by a rule-based system, and therefore we expect these models to do very well on it; (b) supervised models on Hotpot can achieve substantially higher performance as well. (We have added this as footnote 8 in the Appendix)
>
> We will describe the improvements more precisely in any future version.

---

### Official Review · Reviewer_67Ay · 2022-07-07

**Rating:** 4
**Confidence:** 4
**Soundness:** 3 good
**Presentation:** 4 excellent
**Contribution:** 3 good

**Summary:**

This paper aims to investigate different aspects of explanation-enhanced few-shot in-context learning, especially under textual reasoning scenarios with the task of question answering and natural language inference. The paper mainly discusses: 1) whether the explanation fed into the prompt with P-E or E-P format can help the model improve its accuracy on downstream tasks, 2) whether the explanation is consistent with its prediction, 3) whether the explanation itself is factual enough to be trusted, 4) whether we can use the explanation to help the model calibrate its confidence in prediction. The paper performs experiments on three datasets, one synthetic dataset plus two existing annotated datasets. Through the paper, the authors claim that: 1) the explanation can mildly help the accuracy, 2) the explanation is mostly consistent with the prediction, 3) explanations are quite unfaithful, causing significant hallucination, 4) using the explanation to calibrate the model could further help the model.

**Questions:**

Did you also test on the different math-based datasets like MultiArith, GSM8K to evaluate their factuality and consistency, etc?

**Limitations:**

The limitation of this paper is mainly in its deficiency of experiments to support its claims. First of all, the paper only test two tasks, namely question answering and natural language inference. Furthermore, for each of these two tasks, the authors only pick one human-annotated dataset. I think this might not be sufficient to draw a more general conclusion about LLM/GPT-3. I would hope to see more tasks, even many synthesized tasks would help in better understanding LLM's explanations.

The other limitation is that the paper doesn't provide enough detail about its prompt engineering. According to the "Large Language Models are Zero-Shot Reasoners", picking a correct prompt could totally change the landscape. I'm afraid that the lack of comprehensive prompt engineering will make the claims weaker. For the experiment result in Table-1 E-P E-SNLI, I'm not sure whether better prompt engineering or Davinci-02 will make the number totally different.

**Strengths And Weaknesses:**

Strength:
1. the paper builds on top of the recent breakthrough of explanation-based in-context few-shot learning (Chain-of-thoughts) to investigate its reliability issue.
2. the paper investigates a highly important problem in large language model, which could contribute a lot to the community to influence the follow-up research.
3. the paper dives deeper to exploit the issue of explanation and propose explanation-based calibration methods.

Weakness:
(See the Limitation Section)
Without sufficient experiments (synthetic + human-dataset across different tasks), it's hard to draw a more convincing conclusion. On the other hand, re-running the experiments on Davinci-002 will help tease out the confounding factor of prompt/model.

---

> ### Author Response · Authors · 2022-08-02
> **Thanks for the questions; please find answers below**
>
> **Q:** Without sufficient experiments (synthetic + human-dataset across different tasks), it's hard to draw a more convincing conclusion. Re-running the experiments on text-davinci-002 will help tease out the confounding factor of prompt/model.
>
> **A:** Please see the general response. We included additional results on OPT-175, davinci, and text-davinci-002 in Appendix H and Appendix I.
>
> ---
>
> **Q:** The paper doesn't provide enough detail about its prompt engineering. According to the "Large Language Models are Zero-Shot Reasoners", picking a correct prompt could totally change the landscape.
>
> **A:** In our preliminary experiments, we found that GPT-3 Instruct series models are robust with respect to the changes in prompt template (e.g., using “Q”, “A” vs “Question”,“Answer”), and are more sensitive to the changes of “shots” in the prompt. Taking into account that these experiments are expensive, we focus our experimentation on varying the sets of shots in the prompt.
>
> Nevertheless, for the synthetic dataset, we also tried using an alternative style of explanation in our earlier experiments. The performance was slightly worse than the style we used in our main paper. The reliability of the generated explanations in the alternative style is congruent with the findings of our paper. Please see Appendix J for more details.
>
> We acknowledge that including triggers for multi-step reasoning could impact the results. We experimented with adding “let’s think step by step” to E-P, which does not affect the performance. Please see Appendix K for more details.
>
> ---
>
> **Q:** Did you also test on the different math-based datasets like MultiArith, GSM8K to evaluate their factuality and consistency, etc?
>
> **A:** Please see the general response about framing.

---

> > ### Comment · Reviewer_67Ay · 2022-08-07
> > **Great Analysis, raise my score from 4 to 5**
> >
> > I think the new framing of the paper "The Unreliability of Explanations in Few-shot Prompting for Textual Reasoning" makes more sense. Though the mathematical reasoning chain of thoughts are also likely to be unreliable, the paper didn't provide any evidence to show that. Restricting the domain to textual reasoning makes the paper more consistent.
> >
> > Also, thanks for adding comprehensive results for Davici-001 and 002, the results seem very interesting. Davinci-001 and Davinci-002 both use instruction to fine-tune GPT model, but their trends are totally different. Davinci-001 performs very similarly as Davinci and other GPT models, however, Davinci-002 benefits significantly from the explanation in the front. There might be some dark magic behind the scene, but we are not sure what's the actual reason. The author conjectured that this improvements come from data leakage, which I'm not quite convinced because your synthetic dataset also gets huge bump. Anyway, these results seem be consistent with the "Large Language Models are Zero-Shot Reasoners".
> >
> > Overall, I'm satisfied with the author response and would like to raise my score.

---

> > > ### Author Response · Authors · 2022-08-09
> > > **Thanks!**
> > >
> > > Thank you very much for the response! We'd like to respectfully let you know the review score above is not actually updated.

---

### Official Review · Reviewer_NkfS · 2022-07-09

**Rating:** 6
**Confidence:** 3
**Soundness:** 2 fair
**Presentation:** 3 good
**Contribution:** 2 fair

**Summary:**

The authors suggest that conditioning on and generating explanations of a task or problem yields only minimal gains over raw in-context learning, but that the validity of such explanations may be correlated with the accuracy of answers the model produces. They use this hypothesis to motivate approximating reliability automatically as a method of calibrating in-context learning.

Three datasets are used. One is an automatically generated synthetic dataset that is engineered so that automatically testing the reliability of explanations is possible, at least in theory. One is an adversarial subset of a previous QA dataset (HotPotQA) that is balanced so that GPT-3’s default  performance is split 50/50 between correct and incorrect answers. The third is a Natural Language Inference (NLI) dataset with human-annotated explanations.

The authors consider two methods for using explanations: explain-then-predict and predict-then-explain. In the former, the model first provides reasoning and then makes a prediction, with past work making the claim that this explanation helps the model predict more accurately. In the latter, the model predicts then explains, so that its explanation does not directly impact the prediction, but the explanations in the few-shot demonstrations still impact the final prediction. Varying number of few-shot demonstrations are used, as can fit in GPT-3’s context window.

The results show only mild gains from using explanations on these benchmarks (which were specifically intended to probe the use of these explanations and thus may not be representative of the average few-shot learning case). The comparison to previous work is narrativized on lines 123-129, but evidence is not given for whether this narrative holds, which is makes it difficult to assess.

Factuality and consistency are inspectd. For the synthetic dataset, rules are used to judge consistency, but this rule system is not described, even in the appendices. Again, while it is possible these rules are sound, it seems there are many potential pitfalls and not describing these rules makes their validity impossible to assess. For the other datasets, the authors manually annotate 100 of the same examples to report an impressively high annotator agreement. The results suggest that non-factual explanations indicate incorrect prediction, as a rule of thumb. Reporting of results is somewhat inconsistent: on E-SNLI E-P is not reported  and factuality is not judged, though the authors give some reasoning for both of these. Not reporting the E-P strategy because results are so poor still seems strange.

Next the authors attempt to calibrate the model, using a handful of extra data and human judgements. This is especially interesting, as the authors are more-or-less using the full context window available to GPT-3 so these examples could not have been used as in-context demonstrations. A method for calibrating factuality on each dataset is proposed. While sensible, these methods are dataset specific and somewhat adhoc, with no evidence given that they actually correlate with the factuality annotations reported in Table 2.

Despite suggesting that E-SNLI can’t be assessed as factual/non-factual the authors attempt to calibrate E-SNLI using factuality, without discussion of how this works other than using “an analogous score following the same principle” where they consider the premise of the NLI inference as the context. If this is how calibration happens, why couldn’t this have been done when annotating E-SNLI for factuality?

All the calibration methods are shown to improve the accuracy of the results.

Related work is reviewed. A clear discussion of the potential risks of explanations given by models, especially trusting such explanations too much, is presented.

**Questions:**

- Why wasn’t the pseudo-factuality setup used for calibrating E-SNLI factuality also used to annotate E-SNLI?
- What automatic rules were used for analyzing the factuality and consistency of the synthetic dataset?
- Why weren’t any standard few-shot learning datasets considered, especially ones where previous work found explanation generation helpful?

**Limitations:**

~~Authors have adequately discussed the limitations of their work.~~ Edit: Given the new framing, it will be very important for authors to revise the paper to make it clear how the available datasets limit the claims that can be drawn from this work, as the types of textual reasoning studied are rather limited.

**Strengths And Weaknesses:**

Strengths

- The research questions are well-motivated: despite impressive performance on selected datasets it is unclear how much or why explanations help models.
- Evidence given that factuality is correlated with consistency and with non-factuality with incorrect prediction is an interesting result.
- Exploiting extra data that can’t be put into the context window of a model for calibration is potentially a useful technique for many tasks.
- The calibration shows seemingly meaningful performance improvements (though on highly specific and sometimes custom datasets)

Weaknesses

- ~~Only one version of GPT-3 is considered—GPT-3 Instruct, which is finetuned on human feedback, making the results somewhat less comparable to other work.~~ Edit: Authors have added significant new results.
- Result reporting is selective—on E-SNLI factuality is not reported, the E-P method is not assessed,  and Table 3 is missing many values. Edit: Authors have provided reasoning as to why some numbers were not included though in some cases (e.g. the expense of calculating those number), this is still a weakness, albeit a less intense one.
- While the bespoke datasets for looking at this hypothesis are useful tools, it is unclear what these results mean for tasks where few-shot learning is commonly used. Edit: the authors have revises their framing in a way that partially alleviates this, focusing on textual reasoning. However, the available datasets still comprise a tiny fraction of textual reasoning datasets.
- ~~The comparison to previous work is described, but evidence is not directly given. (Lines 123-129)~~ Edit: the authors have specified a new framing that makes this comparison significantly more valid.
- ~~The automatic rules for evaluating factuality and consistency are not described, even in the appendices.~~ Edit: Authors have added a description of their synthetic task.
- ~~Despite suggesting that E-SNLI can’t be assessed as factual/non-factual the authors attempt to calibrate E-SNLI using factuality, without discussion of how this works other than using “an analogous score following the same principle”.~~ Edit: Authors have described why they can heuristically calibrate factuality without being able to assess it.
- ~~The calibration methods are methods are dataset specific and somewhat adhoc, with no evidence given that they actually correlate with the factuality annotations reported in Table 2~~ Edit: Authors have provided reasoning why their calibration methods should correlate with factuality annotations.

---

> ### Author Response · Authors · 2022-08-02
> **Thanks for the questions; please find answers below**
>
> **Q:** What automatic rules were used for analyzing the factuality and consistency of the synthetic dataset?
>
> **A:** Please see Appendix G.
>
> ---
>
> **Q:** Despite suggesting that E-SNLI can’t be assessed as factual/non-factual the authors attempt to calibrate E-SNLI using factuality, without discussion of how this works other than using “an analogous score following the same principle”. Why wasn’t the pseudo-factuality setup used for calibrating E-SNLI factuality also used to annotate E-SNLI?
>
> **A:**
> Lines 223-224 describe how the principle works. The intuition is that a more accurate predicted explanation should at least cover some aspects of the premise instead of being purely based on the hypothesis. This in turn correlates with an explanation more grounded in the input.
>
> However, while this is a rough heuristic that can be useful for calibration, it isn’t suitable to accurately assess factuality. For example, in the second example in Table 11 of Appendix F (inlined here below), the premise and explanations have low lexical overlap, but the explanation is grounded in the premise. Assessing this is highly subjective and requires human judgments, likely multiple judgments by annotators agreeing on some standard. We felt that devising an annotation protocol for it was outside the scope of this paper. Because of the extensive design and vetting process that would be necessary for this, we omitted it to avoid putting forth a human evaluation that may be untrustworthy.
>
> > A person wearing sunglasses, a visor, and a British flag is carrying 6 Heineken bottles.
> >
> > Q: The person has beer bottles and has an england flag and glasses. True, False, or Neither?
> >
> > A: False, because the person could be carrying any number of items and not just six Heineken bottles.
>
> Here is another example of a particularly subjective case:
>
> > A group of rollerskaters skate on a patch of cement with palm trees in the background.
> >
> > Q: A boy is walking around on the grass. True, False, or Neither?
> >
> > A: False, because the boy is not rollerskating.
>
> ---
>
> **Q:** Why is reliability of the E-P method not assessed on E-SNLI?
>
> **A:** We did not evaluate the reliability of E-P for E-SNLI as its performance was far behind Few-Shot and P-E.
>
> ---
>
> **Q:** Why weren’t any standard few-shot learning datasets considered, especially ones where previous work found explanation generation helpful?
>
> **A:** We considered a range of commonly used datasets in the literature (see the Wiegreffe and Marasović survey, 2022). and filtered by the following criteria: (1) textual explanations were annotated in prior work; (2) the datasets focus on primarily textual reasoning (not math reasoning); (3) GPT-3 is not solving the dataset already in a few-shot setup without explanations.
> If the reviewer has suggestions of standard few-shot datasets that satisfy these criteria, we will be happy to consider them for a future version.
>
> ---
>
> **Q:** No evidence given that the scores used for approximating factuality correlate with the actual factuality annotations reported in Table 2.
>
> **A:** We verified the correlations in the following way. Consider using the approximate factuality scores (real values) of explanations generated using the E-P paradigm to predict the ground truth factuality annotations (binary labels). If we treat these scores as a single feature, a classifier based on that feature has an AUC of 76.5, substantially higher than a random baseline of 50.0.

---

> > ### Comment · Reviewer_NkfS · 2022-08-05
> > **Thank you for the clarifications. I'm raising my score from 4 to 6.**
> >
> > Thank you for the very clear answers, the large number of new experiments, and the appropriate reframing of your work.
> >
> > My technical correctness concerns have been alleviated, so I have raised my score significantly, from 4 to 6.
> >
> > I agree that the problem of how much explanations help is deeply important, but I do still worry that the evidence provided here is so limited by the current datasets that it is difficult conclude very much about textual reasoning in general. However, I also agree that this is a necessary first-step.
> >
> > On the subject of what other datasets to use, it seems that the Marasović et al. 2022's FEB benchmark (which you appropriately cite) has three datasets that you do not consider? Is there any reason why those aren't appropriate? Including those would make me, and I am guessing many other readers, more confident your claims are sufficiently general.
> >
> > Marasović, Ana, et al. "Few-shot self-rationalization with natural language prompts." _NAACL Findings_ (2022).

---

> > > ### Author Response · Authors · 2022-08-05
> > > **Thank you very much for the response!**
> > >
> > > **Q:** Have you considered the datasets in the FEB benchmark? Is there any reason why those aren't appropriate?
> > >
> > > **A:** We only became aware of FEB when it was publicized in April 2022 after its acceptance at NAACL. We agree that including more datasets will make the claims more general, but we primarily focused on the most commonly used tasks at the time of our experiments.
> > >
> > > We initially considered CoS-E due to its prominence, but like the FEB authors, we disregarded it due to the poor quality of the gold standard explanations. In fact, as in Wei et al. (2022), simply including explanations in CommonsenseQA only leads to mild improvements, similar to what we reported in our paper. In any case, solving instances in ECQA/CommonsenseQA often involves materializing commonsense knowledge that’s relevant to the question at hand, rather than producing a reasoning chain grounded in a question and some given evidence, which was our focus here.

---

### Author Response · Authors · 2022-08-02
**Response to all reviewers + overview of paper update**

Thanks to all reviewers for the thoughtful comments and feedback! We added substantial content covering new experiments in the appendices of the paper, which we describe below. We’d also like to clarify the main framing of the paper as well as answer some common questions posed by more than one reviewer.

**Changes to the paper**

We added Appendices G-L and left the body of the paper mostly unchanged. We will integrate this information throughout the paper in any future version. The updated paper PDF contains the full appendix for easier viewing, so you do not need to consult the supplementary materials.

The changes are as follows:

* Added the results of few-shot, E-P, and P-E for other language models, including OPT-175B, davinci (GPT-3 API non-Instruct series), and text-davinci-002. Please see Appendix H for details. We also inlined the new table below.
* Added the reliability evaluation for the synthetic dataset on these other language models (Appendix I).
* Added the results of using an alternative style of explanations for the synthetic dataset (Appendix J)
* Added the results of including “let’s think step by step” in the prompt (Appendix K).
* Added the details of automatic rules used for assessing the consistency and factuality of explanations for the synthetic dataset (Appendix G)
* Added information about the cost for running our experiments (Appendix L).

**Framing of the paper**

Our paper focuses on textual reasoning tasks. As a result, we do not extensively discuss symbolic and arithmetic reasoning datasets; some of these bridge textual and mathematical reasoning, but we still consider these problems to have different characteristics. To make it more clear, we changed the title of paper to:

*The Unreliability of Explanations in Few-shot Prompting for Textual Reasoning*

We view our results as complementary to Scratchpad and chain-of-thought: using step by step explanations clearly works well for those types of symbolic reasoning and math problems, and even gives some gains here as we note in the paper. But these methods do not increase accuracy as much as we might expect here, and most surprisingly, the explanations they generate have flaws that we did not expect, most notably factuality. These limitations are, in our view, worth calling out.

**Results using Other LLMs:**

For the revision, we include additional results on OPT-175B, davinci (GPT-3 non-Instruct series API), text-davinci-002 (the latest Instruct series API). The detailed results can be found in Appendix H and Appendix I.
We include the table from the paper; here is our discussion of the results (also copied from the appendix of the revised paper):

| **LM**           | **Approach** | **Synth** | **AdvHotpot** | **ESNLI** |
|------------------|--------------|:---------:|:-------------:|:---------:|
| OPT-175B         | Few-Shot     |  **40.5** |      49.7     |  **44.0** |
| OPT-175B         | E-P          |    29.6   |      **52.6**     |    39.3   |
| OPT-175B         | P-E          |    40.2   |      43.3     |    43.4   |
| davinci          | Few-Shot     |    49.5   |      49.1     |    43.3   |
| davinci          | E-P          |    47.1   |      **54.1**     |    40.4   |
| davinci          | P-E          |    **51.3**   |      48.7     |  **48.7** |
| text-davinci-001 | Few-Shot     |    54.8   |      53.2     |    56.8   |
| text-davinci-001 | E-P          |  **58.5** |    **58.2**   |    41.8   |
| text-davinci-001 | P-E          |    53.6   |      51.5     |  **59.4** |
| text-davinci-002 | Few-Shot     |    72.0   |      77.7     |    69.1   |
| text-davinci-002 | E-P          |  **86.9** |    **82.4**   |  **75.6** |
| text-davinci-002 | P-E          |    81.1   |      77.2     |    69.4   |

---

> ### Author Response · Authors · 2022-08-02
> **Response to all reviewers + overview of paper update (cont'd)**
>
> As shown in the table above, the results on OPT and davinci are consistent with our findings on text-davinci-001.  E-P does consistently provide the strongest performance on the AdvHotpot setting, but the improvements are 5% absolute or less. On Synth and ESNLI, E-P typically degrades performance (except on Synth for text-davinci-001) and P-E is inconsistent across the different models. Overall, vanilla LLMs (OPT and davinci) see limited benefit from using explanations, and even text-davinci-001 does not see substantial improvement.
>
> The only exception is text-davinci-002. text-davinci-002 greatly benefits from explanations in the prompt across all the three tasks, and E-P is consistently more effective than P-E. However, it is unclear what makes the difference. As far as we are aware, the differences between text-davinci-002 and text-davinci-001 are not described in any publication or blog post.(One publicly-described difference is the addition of editing and insertion, discussed at https://openai.com/blog/gpt-3-edit-insert/, but this does not explain the performance differences we observe.)
> Comparing davinci and text-davinci-001, we see the move to Instruct series models is *not* sufficient to explain the difference.
>
> One possibility is that 002 is an updated version of 001 that includes more Instruct data collected using the API. One hypothesis for the improvement is data leakage from our test set. Because we started running experiments for this work in late 2021, it is conceivable that text-davinci-002 was trained on human-written completions for our data. Another hypothesis is that text-davinci-002 features T0-like fine-tuning on some available datasets such as HotpotQA, which would also change the interpretation of the results.
> Given the lack of transparency with this model, we hesitate to make scientific claims here. In any case, the stronger results only appearing in this setting means that we cannot broadly argue that explain-predict is the superior setting.
>
>
> **Missing Numbers in Table 3 and 4:**
>
> The missing numbers either do not apply to those rows or do not affect the claims of the paper.
>
> Line 2 in Table 3: given the limited prompt length, the extra examples cannot be added in the prompt. Naively adding these examples and truncating the prompt would lead to identical results to the leftmost value in this row. So the other values are intentionally left blank.
>
> Line 3 in Table 3: we use the performance obtained using 128 examples as an estimation of the upper-bound performance of Few-shot (NN) (Liu et al., 2022). The performance (58.9) is already lower than any other calibrator-based approaches (FewShot+ProbCalib, P-E+ProbCalib, and  P-E+ExplCalib) using calibrators (even those only using 32 examples). So these values would not affect our claims.
>
> Line 5 in Table 3: Similar to Line 2, they are not applicable given the prompt length limit.
>
> It’s worth noting that filling in Few-Shot (NN) for Table 3 and Table 4 would cost roughly $600. Given that it doesn’t affect the claims, we intentionally avoid doing it to reduce the cost. (We also included information about cost in Appendix L, which is useful information for reproducibility.)

---

### Meta-Review · Area_Chair_urWn · 2022-08-26

**Recommendation:** Accept
**Confidence:** Less certain

**Metareview:**

The authors perform an analysis that suggests that explanations may not provide reliable signal in few-shot in-context learning, showing that adding explanations yields only minimal gains over raw in-context learning. They then develop an approach to approximate the reliability of predictions automatically using these explanations.

In the initial reviews, the reviewers pointed out issues with the empirical rigor of the study and its framing. However, they seem to have addressed these concerns by narrowing the scope of their contributions and providing additional experiments supporting their claims.

**Award:**

No

---

### Decision · Program_Chairs · 2022-09-14

Accept